# RePrompt: Reflection-based Automatic Prompting for LLM Agents

## Abstract

In the past year, large language models (LLMs) have had remarkable success in domains outside the traditional natural language processing, and their capacity is further expanded into the so-called LLM agents when connected with external tools. In all domains, the prompt to the LLMs has been shown to make a big difference in what the LLM would generate and thus affect the performance of the LLM agents. Therefore, automatic prompt engineering (APE) has become an important question for many researchers and users of LLMs. However, previous works in APE rely on a final checker to evaluate the performance of the given prompt – a requirement that is hard to meet in the case of LLM agents, where intermediate feedback is easier to obtain, and the final evaluation could be expensive, inaccurate, or even missing. In this paper, we focus on test-time prompt optimization, where such a solution checker is not available. We propose a novel method, REPROMPT, which does a "gradient descent"-like approach to optimize the step-by-step instructions in the prompts given to LLM agents, based on the chat history obtained from interactions and reflections with LLM agents. By leveraging intermediate feedback, REPROMPT can optimize the prompt without the need for a final solution checker. We evaluate our approach on PDDL generation, TravelPlanner, and Meeting Planning to show that our method could generally improve performance for different reasoning tasks and on different models.

## 1 Introduction

Large language models (LLMs) have won significant success since the release of ChatGPT (OpenAI, 2022). In addition to traditional natural language tasks like summarization and sentiment analysis, LLMs have been shown to be effective in many domains that are closer to applications like code generation (Chen et al., 2023; Roziere et al., 2023), human-computer interaction (Li et al., 2023) and math problem solving (Wei et al., 2022; Yu et al., 2024). While pure LLMs are limited in their reasoning capability (Sun et al., 2023; Valmeekam et al., 2023; Chen et al., 2024a), researchers have introduced tool-use to LLMs and built integrated systems, namely *LLM agents*, to enable the use of LLM in even more general domains (Wang et al., 2023a; Mao* et al., 2023; Xie et al., 2024).

Prompts play a crucial role in these successes, as variations in prompt design can lead to dramatically different success rates (Wei et al., 2022). Consequently, prompt engineering is often necessary for task-specific optimization. However, manual prompt engineering is both challenging and time intensive, motivating the development of automatic prompt engineering (APE), where LLMs generate prompts in their preferred natural language (Zhou et al., 2023). With some trials, APE can efficiently converge to a robust prompt, outperforming the original prompts that are often simple (Zhou et al., 2023; Zhang et al., 2023).

However, in complex LLM-agent tasks such as reasoning, APE is still under-studied, and most users still use primitive prompts or carefully hand-crafted prompts in their LLM. On the one hand, this is because LLM agents normally have high constraints on the output format, and previous APE methods can easily break the format requirement. On the other hand, existing APE methods rely highly on trying different prompts, checking the performance of each one, and finding the best one. However, there are numerous cases where an accurate final evaluator is highly costly, making it impractical for most people to use the

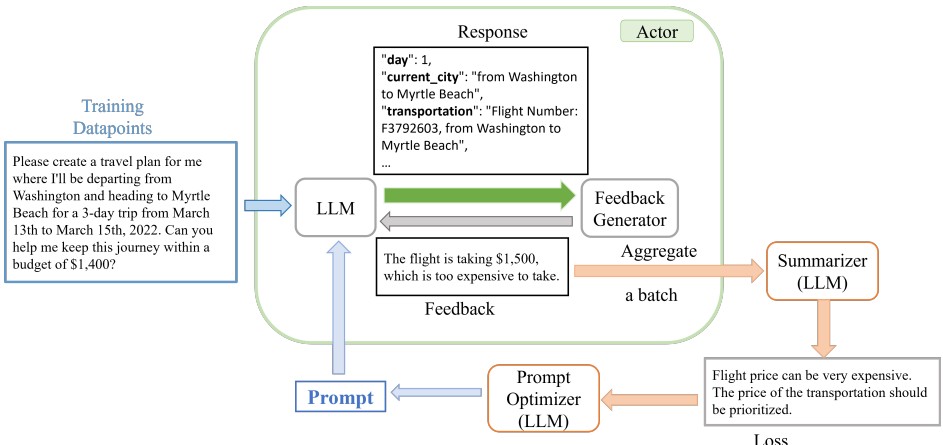

Figure 1: The workflow of our method REPROMPT.
.

evaluator frequently during training or, in our case, while optimizing the prompt. This is especially common in high-specialization domains that demand extensive domain knowledge, such as physics, chemistry, and other scientific disciplines. There are also scenarios where a final evaluation might be entirely absent, as seen in applications like ChatGPT, and particularly in GPT's tools. In these situations, users interact with the application, sometimes providing intermediate feedback, but often leaving without clarifying whether they received a satisfactory response or abandoning the interaction due to dissatisfaction with the model's ability to assist further. Without cheap prompt performance evaluators, existing APE methods cannot be applied to these scenarios.

In this paper, we focus on a scenario where there is a specific reasoning task one wants to use LLMs for, while there is no ground-truth checker to check the correctness of the outputs. One example is the OpenAI GPTs tool in ChatGPT (OpenAI, 2023b) to plan their travel or help in writing a code. In these domains, LLMs typically use a Chain-of-Thought (CoT) prompt with interactive procedures like REACT (Yao et al., 2023b) and REFLEXION (Shinn et al., 2023) to improve their performance. We formally define the problem as test-time prompt optimization, and propose a novel automatic prompt engineering method called REPROMPT, which takes the common practices of using CoT and REACT into consideration and uses the dialogue history from these results as the information for each prompt update. By summarizing the dialogue history and then analyzing how to improve the prompt step-by-step, we optimize the prompt based on past history while not overfitting to corner cases. An overview of our proposed framework is shown in Figure 1.

We benchmark our approach on Planning Domain Definition Language (PDDL) generation (Guan et al., 2023), TravelPlanner (Xie et al., 2024), and Meeting Planning Zheng et al. (2024) to show that our methods can achieve a higher first-round success rate, and our methods can be combined with different types of feedback generator.

In conclusion, our contributions are:

1. Propose the problem of test-time prompt optimization.

2. Propose to use "gradient-based"-like prompt optimization in LLM agents.

3. Propose a summarization-based prompt optimization that focuses on optimizing steps in the prompt, and demonstrate that optimizing the steps is an efficient way of prompt optimization in LLM agents.

4. Our proposed method does not require a solution checker, and can be used in LLM-agents scenarios where such a checker is not available.

## 2 Related Works

Our work lies at the intersection of prompt optimization and LLM for reasoning.

In prompt optimization, many works have proposed to optimize the prompt using differentiable tuning on soft prompts (Lester et al., 2021; Qin & Eisner, 2021), train auxiliary models as the optimizer (Hao et al., 2023; Deng et al., 2022; Zhou et al., 2023), or directly train the prompter themselves (Wang et al., 2023b). This line of work requires access to the model weights of the language models and is not generally applicable in the current era of using LLMs like GPT-4 (OpenAI, 2023a) and Claude-3 (Anthropic, 2024) through APIs. Another line of work chooses to use machine learning models to provide approximate guidance on which prompt is better, either by using reinforcement learning (Shin et al., 2020; Zhang et al., 2023; Chen et al., 2024b) or by discrete manipulation with LLM feedbacks (Guo et al., 2023). There are also some other works in prompt optimization that propose a relatively general solution, such as using beam search or Monte Carlo tree search as the "gradient descent" optimizer (Pryzant et al., 2023; Tang et al., 2024; Wang et al., 2024). Another distinctive work explores iterating prompts for individual test cases by progressively inserting improved solutions as examples Yang et al. (2023). Our work is very close to the second group of works, which focus on doing "gradient descent"-based prompt optimization, and can be seen as a generalization of the current methods to reasoning domains. In contrast to prior works, this paper focuses on the setting of test-time optimization, which does not rely on ground-truth feedback to evaluate the performance of the current prompt.

In LLM for reasoning, a key challenge identified by researchers is how to correctly use prompts to guide the LLM to generate useful auxiliary output that leads to a good final solution. Chain-of-Thought (CoT) (Wei et al., 2022) is the most commonly used prompt that can improve the performance of LLMs that consists of simply adding a fixed sentence, such as "Let's think step by step.". More recently, Tree-of-thought (Yao et al., 2023a) and Graph-of-Thoughts (Besta et al., 2024) were also proposed as an extension of the simple line-based architecture of auxiliary output to tree and graph-structured output. Orthogonally, researchers have also found that utilizing the interaction capability of LLMs could also improve their performance. Yao et al. (2023b) proposed REACT by letting the LLMs list some thoughts before proposing the actual actions. REFLEXION (Shinn et al., 2023) prompts an LLM agent to reflect on the actions, and save the reflections in the memory to improve efficiency. There are extensive other works like self-refine (Madaan et al., 2023), RCI (Kim et al., 2023), and self-debugging (Chen et al., 2023) that use similar ideas of providing feedback as guidance in the reasoning-related task and strategically adapt the idea to fit the needs of specific domains. Furthermore, recently, o1-like models Jaech et al. (2024) naturally integrate this self-correction process into the generation procedure. Our approach uses this iterative workflow that incorporates feedback before finalizing the answer, optimizing the prompt based on the provided interaction history. This process mitigates early-stage errors and, more importantly, reduces the inherent randomness seen in methods like REACT and REFLEXION. By summarizing recurring issues — such as budget balancing — our method, REPRMOPT, ensures these concerns are systematically addressed within the prompt. This allows REACT and REFLEXION to focus on case-specific challenges, such as unexpected price surges for hotels in particular cities on specific days in the case of solving a TravelPlanner task.

## 3 Methods

### 3.1 Background on LLM agents

In this paper, we consider the problem of LLM agents for reasoning tasks. For LLM agents to focus on reasoning, the Chain-of-Thoughts (CoT) Wei et al. (2022) is one of the most popular methods used in LLMs. By adding a simple sentence of "Let's do it step-by-step," LLMs will automatically outputting auxiliary steps before generating the final answer. Based on this, there has also been recent success in OpenAI o1 Jaech et al. (2024) and Deepseek-R1 et al. (2025) which further demonstrate that by introducing more steps before the answer, LLMs can solve harder problems.

Because LLM might not be correct in the first shot, prior approaches have proposed allowing a few inter-actions before the final answer is given by the LLM (Yao et al., 2023b; Shinn et al., 2023). In these cases,

users provide the LLMs with some feedback and let the LLMs try again with the additional information. This information does not necessarily provide any hint about the final solution, but can be an error message about why the current solution is not correct. For example, this information could be a Python runtime error message in code generation tasks. How to use specific prompts with error messages would mainly depend on each specific task. For example, in the widely used REACT (Yao et al., 2023b), LLMs are required to provide thoughts on the current results before doing the next round, and these thoughts are often not a concrete error but may also include a short analysis that reaches a conclusion that the current action is good. However, due to the difficulty of LLM agent's tasks, checking how good a given prompt is could be expensive and unsuitable for regular in the prompt optimization process. As we mentioned earlier, this could be either very expensive in domains that require high intelligence like physics, or completely infeasible in applications like the web version of GPTs.

## 3.2 Test-time Prompt Optimization

In this paper, we address the new problem of test-time prompt optimization. Unlike prior approaches that optimize prompts a priori using labeled data, our method is designed to operate at test time, when ground-truth labels are not available. This key difference distinguishes our work from existing methods such as PromptAgent (Wang et al., 2024) and TextGrad (Yuksekgonul et al., 2025), which depend on ground-truth checkers to identify reasons for success or failure and to optimize prompts accordingly. Although an LLM-based judge (Chiang et al., 2024) can serve as an approximate checker, the accuracy gap introduced by this substitution is significant. This discrepancy in supervision quality can lead to performance differences similar to those observed between reinforcement learning with verifiable rewards (RLVR) (et al., 2025) and reinforcement learning from human feedback (RLHF) (OpenAI, 2022). We include the comparison to LLM-as-a-judge as a baseline in our travel-planning experiments.

We argue that this new setting better reflects practical applications of prompting for LLM agents, such as in GPTs (OpenAI, 2023b), and in highly technical domains like physics and chemistry. In the next section, we introduce our algorithm, REPROMPT, which, although similar to prior work at a high level, incorporates essential modifications to accommodate the unique challenges of this test-time setting.

## 3.3 RePrompt

### 3.3.1 Overview

In LLM agents for reasoning tasks, we consider the task planning part of LLM agents with prompt optimization, where the tasks of the agents are known ahead of time, and the final solution checker is not available due to its cost.

As shown in Fig. 1, our method, REPROMPT, is a prompt optimizer that is based on the interaction-based action-generating process. Our prompt optimization method is similar to a machine learning training loop, which iterates between getting the output based on the current parameters, calculating the loss based on the output, and optimizing the parameters based on the loss. But in our case, the parameters to be trained are the prompts to be fed into the LLM model, the model forward pass is replaced by the complete interaction-based action-generating process, which includes the feedback generator, and final components of REPROMPT are the loss and optimizer, which are both LLMs instead of numerical calculation for the distance and the gradients.

Given a specific small dataset of reasoning tasks used for training, we first let the LLMs generate their responses using the current prompt. This process needs to include some interaction schemes with some kind of feedback generator like REACT or REFLEXION, but we do not put any constraint on how this part should be done, or how accurate the feedback is. We call this process the *Actor*.

We then wait until a complete batch of chat histories has been collected, at which point we input the entire batch into another large language model, which we call the *summarizer*, to summarize the primary focus point. This focus point might be a recurring issue that frequently prolongs iterations or specific suggestions (or "thoughts" in the case of REACT) that have proven effective in producing high-quality responses. Typ-

ically, the essential information is already present in the chat history and does not require further analysis or summarization. Our summarizer is designed to capture key insights across various scenarios, omitting scenario-specific details and recommendations while avoiding overly broad summaries that would demand additional reasoning steps. Due to space constraints, we provide the prompt for this summarizer in Appendix E. Unlike prior works, which rely on ground-truth checkers to separate successful from unsuccessful trials before comparison, our approach aggregates feedback from all trials without differentiation, using only the signals produced by the *Actor*. Without explicit labels, identifying informative cues becomes substantially more challenging, and naively leveraging the raw aggregate often proves ineffective.

With the summarized typical errors identified, we leverage another LLM, the *Prompt Optimizer*, to refine the prompt accordingly. After the steps, we will get an updated prompt, and we can continue to do more iterations, which can also be seen as more epochs as an analogy to training ML models, until the prompt has been sufficiently updated. This updated prompt will help improve the generated result in the first round, and also help ensure common problems that can be fixed by the feedback generator are resolved as early as possible. During test time, we directly use the updated prompt and test it on the new test set. During later tests, we do not require the exact same process to generate the response, e.g., the feedback generator can be removed completely from the *Actor* procedure if it is quite expensive.

### 3.3.2 Optimizer

Because the *Actor* can adopt any architecture and the *Summarizer* merely aggregates the collected feedback, we direct our attention to the *Prompt Optimizer*. For this component we propose a novel, step-based design that tackles the central challenges faced by LLM-driven agents.

To begin with, this optimizer LLM is instructed to adhere to the following principles that are commonly used in prompt engineering:

1. The refinement process should prioritize the common structural components of the prompt rather than scenario-specific elements that vary across data points. For instance, in the task of PDDL formulation generation, a frequent suggestion is to include additional details about the specific domain the LLM is targeting, along with more comprehensive background knowledge. However, since our goal is to construct a generalizable prompt capable of handling diverse PDDL formulation tasks, incorporating such domain-specific details would reduce its applicability and should therefore be avoided.

2. The improvement should prefer to identify whether the specific problem does occur in the given scenario. For example, suppose there is a certain budget one wants the solution provided by LLMs to satisfy, and the previous history shows that this budget constraint could be one of the main problems that lead to a wrong solution. In that case, the cost of a typical plan should first be approximated. If it breaks the constraint, it could prioritize the budget constraint when getting a solution; otherwise, it should ignore this problem.

Based on the above principles, we use the prompt to ask the *optimizer LLM* to do the following step-by-step, as detailed in Appendix E:

1. Propose a few potential solutions to the problem.

2. Analyze the solutions one by one to see which one meets the rules better.

3. Choose the single solution that is the best. Unlike some of the existing work (Zhou et al., 2023; Deng et al., 2022), we do not ask the LLM to give a concrete number as the value of the prompt sentence.

4. Analyze the original steps in the original prompts, check whether the chosen solution should be inserted before the current step or the solution is a more concrete detail on the step, and the prompt on the current step should be replaced by the solution. If it is the step, add the prompt here.

5. Output the final prompt that combines the original prompt and the updated prompt.

To assist the optimizer in handling common challenges, we have pre-encoded several frequently used solutions directly within the optimizer prompt. While these solutions could be discovered over multiple iterations, providing them upfront minimizes the number of iterations required for prompt refinement, accelerating the optimization process.

Furthermore, while we understand that in-context learning is very important for reasoning, we found it extremely challenging to update the examples to follow our instructions step-by-step perfectly all the time. And therefore, we choose not to change the examples at all. In most cases, the examples serve as a hint to the LLM on the output format and the related capabilities rather than a concrete guide on how to follow the step-by-step instructions, and we currently do not see any empirical drawback by not updating them.

### 3.3.3  Prompt Intialization

To initialize the training process, we start with the original prompt. Because not all original prompts include detailed step-by-step instructions that our optimizer can leverage, we introduce an auxiliary checker (excluded from the main workflow for simplicity) to convert prompts into a structured, step-by-step format when needed. Specifically, we first use a language model to assess whether the current prompt already contains such instructions. If it does not, we manually append a standardized sequence of steps to the prompt, placing it just before any examples. This sequence comprises two primary steps: a brief problem analysis followed by the solution. This structure is functionally similar to a single "Chain of Thought" (CoT) prompt (Wei et al., 2022), "let's think step-by-step" in most reasoning tasks by encouraging essential analysis without introducing domain-specific knowledge, and also the recent paradigm of "thinking" in reasoning models Jaech et al. (2024).

### 3.3.4  Final Prompt

Note that in the optimization process, REPROMPT only changes the step-by-step instruction phase rather than any other problem description or format requirement specified in the prompt. This brings us to, in general, three possible formats of prompts that the algorithm will end up with:

1. If the current prompt is ReAct-like, or o1-like Jaech et al. (2024), which already includes a step-by-step instruction that gives a specific step, like Thought in REACT, to include all the potential analysis, our prompt will converge to always update this thinking step by adding more and more hints on what to do to it. Our method becomes an algorithm that gives a more specific hint on what part the analysis should focus on compared to other prompt engineering work that introduces hints dynamically.

2. For step-by-step prompts, such as those involving mathematical or logical problem-solving, our algorithm progressively refines the generated procedure by incrementally adding steps. This iterative expansion enhances the concreteness of the planning process, providing a more structured pathway for the model to explore. By guiding the LLM toward the correct final answer, our approach effectively decomposes high-level tasks into granular substeps, improving both reasoning accuracy and interoperability.

3. In some rare case, a list of original guidance is already provided. While they do not require the *Actor* to follow them in order, this can sometimes also be seen and recognized as step guidance. In this case, REPROMPT will adjust this part or maybe add some additional guidance. But overall this is a very rare case.

None of the three prompt formats is universally superior; their effectiveness depends on the underlying task. As an APE algorithm that relies on LLM to optimize, REPROMPT integrates seamlessly with all three settings. Detailed results and illustrative examples are presented in the experimental section and Appendix.

| | # of Incorrect Actions | | | Total # of Errors | | |
|---|---|---|---|---|---|---|
| | TyreWorld | Logistics | Household | TyreWorld | Logistics | Household |
| Dataset Size | 13 | 6 | 22 | INF | INF | INF |
| Guan et al. (2023) | 3 | 1 | 19 | 6 | 1 | 52 |
| REPROMPT | 3 | **0** | **12** | **4** | **0** | **23** |

Table 1: The results on generating PDDL instances correctly without any additional domain expert help. The number of actions is the number of tests provided to the LLMs, and each action can have as many errors as it wants, annotated by human experts. Our method, REPROMPT, is trained for only 1-epoch with only the annotations used to evaluate the original results, and without additional annotation from human experts in training.

## 4 Experiments

### 4.1 Experiment Settings

To test the capability of our algorithm in different scenarios, we choose three environments, PDDL generation (Guan et al., 2023), TravelPlanner (Xie et al., 2024), and Meeting Planning from Natural Plan Zheng et al. (2024). The first two tasks are selected because their feedback generator is already included in the paper, and the reality is that both datasets are hard to conduct iterations with feedback generators. The Meeting Planning task is selected as another LLM agent task that differs from the existing ones.

The PDDL generation task provides accurate but expensive feedback and challenges the exact translation capability of the LLMs, which is necessary for LLMs to be further able to write correct code. The TravelPlanner environment, on the other hand, provides cheap but not accurate feedback through Reflexion without knowing ground-truth information. TravelPlanner also provides tools to be used to query the cost information in the database and challenges the reasoning capability by asking for direct generating solutions. Furthermore, in TravelPlanner, we are testing REPROMPT with REFLEXION, which further includes the thought-action-observation steps rather than a standard step-by-step instruction in the PDDL where each step is an intermediate step that could be helpful for guiding the generation for final results. For Natural Plan, we test the performance of deepseek-R1 et al. (2025), an open-source model that provides its complete long chain-of-thought, and we directly use its own thinking part in the output as the feedback as recent models like deepseek-R1 already included self-evaluation and self-correction in its outputs. Given the different types of feedback, the purpose of our REPROMPT also change: in PDDL, our REPROMPT serves to improve the generation performance without any iteration between the LLM actor and the feedback generator that is used to reduce the cost of generating feedback; in TravelPlanner, REPROMPT is used to help guide the LLM to take all the important steps of concern in all scenarios and reduce the potential failures; in Meeting Planning, REPROMPT is used to reduce the unnecessary trail happens in the thinking process of the LLM, and also reduce the randomness on the steps in the thoughts.

Since the setting of test-time prompt optimization has not been previously explored, and existing prompt optimization methods are not directly applicable, we primarily compare REPROMPT with the prompts provided in the original dataset. Our goal is to demonstrate that REPROMPT can generate more effective prompts in the test-time setting.

In the experiments, the stopping criteria and other hyperparameter settings for each domain will be the same as in the original environments. For PDDL generation and TravelPlanner, we use a temperature = 0, a seed of 42, and results are tested on GPT-4-turbo-1106-preview. To help reproducibility, we provide all the optimized prompts generated by REPROMPT in Appendix F.

### 4.2 PDDL Generation

We first test the Planning Domain Definition Language (PDDL) generation task (Guan et al., 2023). Given a natural language description of a PDDL instance, the job is to define the precondition and the effects of the

actions in PDDL. Specifically, we consider the very first step of constructing the model and do not further consider the later correction phase and PDDL translation phase. [1] After generating the preconditions and effects of the actions, human domain experts are introduced to check whether the generation is correct. In this paper, we incorporate both human domain experts and a separate LLM-based checker to evaluate our results. Specifically, the LLM-based checker is used to verify whether any of the errors identified in the outputs of the original method (Guan et al., 2023), which were released alongside its code, also appear in our results. In addition, the human experts are asked to identify any further errors beyond those detected by the checker.

In this experiment, we use the generated result and the annotation from human experts of the prompt from the previous paper Guan et al. (2023) in "Tyreworld" as the chat history used in the REPROMPT training set, and update it for one epoch to get the updated prompt. Here, because the feedback is provided by domain experts, it is accurate but expensive, so multiple rounds of iterations are not feasible, and so we choose to only train REPROMPT for one epoch and greatly reduce the need for extra annotations. As shown in Table. 1, the prompt we get from REPROMPT not only outperforms in the set that we are training on, i.e., the Tyreworld domain, but also generalizes to other related domains and improves the success rate there. Interestingly, we found that after changing the prompt, the prompt does not introduce any new errors, i.e., the errors the new prompt made are a subset of the errors made by the prompt in the original paper. With this subset of errors, fewer domain experts will be needed to give annotations, and make the whole PDDL translation process much faster.

Among the remaining errors, some stem from the omission of common knowledge explicitly stated in the description. For instance, in the action "Empty a Vacuum Cleaner," the description includes the sentence: "The trash can should be opened if it's openable." While this is a commonsense statement with no new information, in the given context, it implicitly defines a necessary precondition. However, the LLM-generated PDDL precondition fails to capture this, leading to errors. Similar issues occur multiple times, collectively accounting for a significant portion of the observed errors, which can be categorized under this broader pattern.

### 4.3 TravelPlanner

Next, we test on the sole-planning setting in TravelPlanner benchmark (Xie et al., 2024). In this benchmark, the LLMs are required to provide a concrete day-to-day plan, including where they should stay, eat, and how they should travel, and satisfying both commonsense constraints like reasonable city routes and budget constraints. While there are some breakdowns of what specific kind of constraints the plan does not satisfy, the primary metric that is used for comparing different methods is the final pass rate. It needs to be addressed that in this benchmark, the evaluation is done after the act loop is done, separately with a ground-truth checker rather than directly to the feedback loop in REFLEXION, and thus, the feedback in the chat history used by REPROMPT does not actually involve any human interference or oracles on what is the correct answer and what is the list of constraints. This allows us to train on a subset of the test set without worrying about leaking any extra oracle information to the model. Because of this, we choose to report results on the validation set of 180 data points instead of results on the larger test set in order to save API costs. And because REPROMPT is based on further collecting data, we choose a subset of 10 data points in the validation set as our training dataset. We report the same group of metrics defined in the original paper.

As shown in Table. 2, the prompt generated by 5 epochs of training of REPROMPT is better than the pure REFLEXION Shinn et al. (2023) result in the final pass rate, and also outperform PromptAgent Wang et al. (2024) as an example of previous automatic prompt engineering work that was primarily designed for single-round question-answering task [2]. In the optimization procedure, unlike the PDDL environment, the optimized prompt after one epoch does not show any benefit. This is because, as we discussed earlier, the prompt after the first epoch will only include one additional suggestion, which is about looking into the budget constraint in this case. No matter what this round of updates is, it is something summarized

---

[1]At the time of submission of our paper, the evaluation phase is missing in the official Github repository, and we are not able to compare the success rate in those phases in a fair manner.

[2]Because their design was not suitable for Travelplanner, we have made necessary adaptations to make it testable in this case. We provide more details in Appendix B.

| | Delivery Rate | Commonsense Pass Rate | | Hard Constraint Pass Rate | | Final Pass Rate | Train Set Final Pass Rate |
|---|---|---|---|---|---|---|---|
| | | Micro | Macro | Micro | Macro | | |
| Reflexion | 76.67% | 56.39% | 3.89% | 37.39% | **33.89%** | 2.78% | 1/10 |
| PromptAgent | 94.44% | 56.39% | 3.89% | 32.61% | 30.22% | 2.11% | **2/10** |
| RePrompt (1-epoch) | 89.44% | 64.03% | 3.89% | 35.0% | 32.78% | 2.78% | 1/10 |
| RePrompt (5-epochs) | **99.44%** | **80.00%** | **6.11%** | **48.81%** | 25.56% | **3.89%** | **2/10** |

Table 2: Results on TravelPlanner Benchmark. The best results are marked in bold. The delivery rate, commonsense pass rate, and hard constraint pass rate overall contribute to the final pass rate, which is the main metric in this table. Because we are "training" on part of the data, but not using any additional ground-truth information, we split the results into overall final pass rate and training set final pass rate. All the metrics in the table are better when the numbers are larger.

| # Training Samples | Final Pass Rate |
|---|---|
| 1 | 1.22 |
| 2 | 2.44 |
| 5 | 3.44 |
| 10 | 3.89 |
| 25 | 2.00 |

Table 3: Number of Training Samples and Final Pass Rate for RePrompt on TravelPlanner.

from the thoughts provided by the REACT scheme and something the iteration loop of generating a final plan has often already noticed and addressed. The optimized prompt after 5 epochs helps LLMs perform better in both the dataset we used for training and the other data not included in the optimization process. This shows the generalizability of the prompt we get through the process. We observe that our baseline, PromptAgent, tends to change an extensive amount on the original prompt, and becomes even worse on the final test. Additional results using GPT-4o, along with relevant implementation details, are provided in the appendix.

To better understand the source of our improvement in final pass rate, we analyzed its impact on macro commonsense pass rate—a key bottleneck in leveraging LLMs for TravelPlanner. As shown in Table 2, our method substantially enhances performance in this metric. Further analysis reveals that our prompt significantly improves the pass rate for the so-called "reasonable city route" constraint, a critical aspect tested by TravelPlanner (though not explicitly shown in Table 2). This common sense is to ensure "Changes in cities during the trip must be reasonable". This represents a commonsense constraint that LLMs can recognize during the process. However, while such constraints occasionally appear in the feedback loop's reasoning, they are not consistently addressed. Our REPROMPT framework successfully identifies and integrates this constraint into the prompt, ensuring that it is accounted for in most iterations. We believe this is one example that our algorithm has addressed the challenge of "Agents struggle to align their actions with their reasoning." mentioned in the original paper Xie et al. (2024). However, for our baseline PromptAgent, the extensive changes have made the LLMs fail to follow the desired format, leading to more useless steps and more failures.

Surprisingly, our method does not mitigate the issue of agents producing hallucinations due to information confusion. Specifically, our agents continue to generate incorrect flight numbers—assigning them to the wrong flight leg or erroneously using the same flight number for both departure and return flights. While, in theory, such errors could be corrected by the feedback generator, we observe that this feedback is rarely generated or incorporated into the prompt optimization loop. Moreover, our REPROMPT training loop fails to detect these errors. Moving forward, we aim to explore how our REPROMPT framework can serve as an additional verification layer for simple hallucination errors. One potential approach is incorporating scenario-specific information and chat history when computing the "loss."

| # Prompt | Final Pass Rate (%) |
|----------|---------------------|
| Original | 21.00 |
| REPROMPT | **26.00** |

Table 4: Results on Meeting Planning from Natural Plan benchmark. Results are based on Deepseek-R1.

**Ablation Study**  Next, we provide an ablation study on the sampling size and the number of iterations. To fix the total budget spent on training the same, we fix a multiplication of the number of training samples times training epoch to 50. So, with more training samples, one will have fewer training epochs, which might lead to returning a prompt before its convergence. As a special case, when training samples are equal to 1, our method is a vanilla prompt optimizer without a summarizer that is done per scenario. We show our results in Table. 3. We found that selecting an appropriate number of training samples—balancing diversity with the number of epochs- can significantly enhance the performance of REPROMPT. This aligns with our intuition, as REPROMPT can be viewed as a form of gradient descent.

## 4.4  Meeting Planning

Lastly, we evaluate our algorithm on the Meeting Planning task, which involves scheduling meetings with friends while considering availability and travel-time constraints, aiming to maximize the total number of successful meetings. A key challenge of this task is that not every meeting can be scheduled for every problem instance, making it impossible to determine whether an optimal solution has been achieved. In real-world applications, this uncertainty complicates the feedback loop, as accurate assessments of solution quality are inherently difficult. Our approach leverages the think section of DeepSeek-R1, presenting a unique challenge for REPROMPT: effectively extracting useful insights from long and unstructured chat histories. To ensure compatibility with the R1 model, we made specific modifications to the dataset, detailed in Appendix B. For training, we use the first nine data points, applying a batch size of three, and further evaluate performance on the first 100 data points. [3]

The results, presented in Table 4, show that despite the presence of highly noisy thinking logs, REPROMPT still improves the original prompt. This demonstrates REPROMPT's adaptability to the latest models, highlighting its ability to improve performance without relying on a specific type of feedback information.

## 5  Error Analysis

In our experiments, our automatic prompt optimization process does not guarantee the successful generation of an improved prompt. In this section, we outline common errors that arise and describe the ad-hoc strategies we employ to mitigate them. While these methods are neither universally necessary nor broadly applicable across domains, we include them here as pragmatic solutions to specific issues. We anticipate that as LLMs' instruction-following capabilities continue to advance, these ad-hoc interventions will become obsolete and can be seamlessly eliminated.

### 5.1  Incomplete Prompt

The prompt optimizer occasionally fails to generate a complete prompt. As detailed in Appendix E, we have incorporated explicit instructions to ensure completeness; however, as shown in Fig. 2, LLMs sometimes produce prompt templates that require manual copy-pasting by users to finalize the prompt. This issue arises in both our algorithm, REPROMPT, and our baseline, PromptAgent. We observed that this failure is more frequent when the initial prompt is relatively long, likely because LLMs are trained to generate concise responses whenever possible, despite our instructions to output a fully formed prompt. To address this, we introduce an additional LLM to automatically complete the prompt template. This approach enables us to generate fully structured prompts in the TravelPlanner domain. We opted against a rule-based fixer, as the

---

[3]We do not use the complete dataset mainly due to the limited availability of the Deepseek-R1 when the experiment is conducted.

> You are defining the preconditions and effects (represented in PDDL format) of an AI agent's actions. Information about the AI agent will be provided `...`
> Before defining the preconditions for an action, consider the implications of the action within the given domain.
> Here are two examples from the classical BlocksWorld domain to demonstrate the output format.
> \<Examples from the original prompt\>
> Here is the task.
> Domain information: {domain_desc}
> Action:

Figure 2: An example output of the optimizer LLM that outputs a prompt template instead of a complete prompt. While it is technically correct and successfully added the additional instruction shown in blue, this output is not acceptable since it includes a template holder for examples marked in red, and this output still needs post-processing with the original prompt to complete the prompt. For simplicity, we have omitted the parts of the original prompt that are not changed, marked in green, and this part of the prompt can be found in the original paper (Guan et al., 2023).

generated templates use a variety of delimiters—including but not limited to <> and —making it impractical to manually define exhaustive replacement rules. Instead, we rely on the LLM's ability to recognize and complete these templates autonomously, reducing manual intervention and improving efficiency.

### 5.2 Incorrect Change by Accident

In some domains, the output format can be similar to a more commonly used domain, and LLMs are misled to correct the prompt in certain parts. For example, in our PDDL domains, we ask the LLM to generate the preconditions of the actions rather than the actual PDDL file. In our experiments, we found that even though our prompt has explicitly required the LLM not to change the output format part of the prompt text, the updated prompt still sometimes changes the output format by mistake, specifically, changing the output of "Preconditions" in capital into "precondition" in smaller cases. To solve this problem, we leverage the feedback of the syntax checker. While the generated results could have some errors, the results should always be complete, and have the required syntax. And if our syntax parser that extracts the answer from the LLM output cannot find the word "Precondition", we know the prompt used is not correct, then we re-run REPROMPT on the same step to generate a correct one. Because the fail rate with our current code is empirically less than 10%, this ad-hoc solution is enough.

## 6 Conclusion

In this paper, we have focused on optimizing the prompts used in LLM agents. We propose a new automatic prompt optimizer, REPROMPT, which is based on the summarization of the interaction between LLM agents and feedback generators and optimizes the step instructions in the prompt. Although the performance is still limited by the natural capability of the feedback generation loop, our experiments across three challenging reasoning tasks show that LLM agents could benefit from an updated prompt, regardless of the type of feedback provided and without the need for a ground-truth checker to be included in the procedure.

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

# A    Limitations

It is important to discuss the limitations of our proposed method REPROMPT. First of all, as the prompt is optimized similarly to fine-tuning, our generated prompts are also limited to the training data and harm the generalizability of LLMs to a certain degree, i.e., if the training data demonstrate unique challenges to LLMs that only occur in those training scenarios but not general cases, our optimized prompts may even be less efficient compared to the original prompts.

Next, our prompts rely on comprehensive tools available to the LLM agents. Because the optimized methods are provided directly from LLMs rather than processed with a search-based method, REPROMPT can propose to use some statistic tools that are not available in the actual given settings. We leave the possibility of letting LLM code the extra, not-available, but commonly used tools themselves to future works.

Furthermore, sometimes, the feedback generator, which we do not have and put any control on, can generate useless or even wrong and misleading results. While REPROMPT is based on summarization, REPROMPT will take such feedback into the prompt if such a mistake is often made. And because we do not consider removing useless steps in the prompt in this paper, such a mistake will only increase the total number of tokens used, not contributing to better results. Future work could add a search-based mechanism to identify such a mistake and potentially fix it, but this will potentially require more ground-truth feedback from the environment and can lead to a more constraint in applicable domains.

Last but not least, our proposed method is doing the planning in the prompt phase, and thus, if the LLM agents are proposed for very general domains that need completely different procedures in different scenarios, e.g., LLM agents for solving math problems, our proposed method will not work at all. However, if the LLM agents are proposed for specific tasks, like using LLM agents to solve high-school geometry problems, our proposed method could help learn the planning very efficiently, as shown above in the experiments.

# B    Implementation Details

## B.1    PromptAgent Baseline

PromptAgent Wang et al. (2024) was initially proposed as an APE algorithm for question-answering tasks, which made its design mostly use the fact of ground-truth feedback generator which tells at least whether the final prediction is correct or not. In our TravelPlanner settings, the environment does not even provide this boolean checker during the APE process.

To adapt the PromptAgent framework to our settings, we created an LLM-based checker, which provided exactly the same input as the plan generator and the plan just generated. The checker is called after the reflection is finished and submitted a plan, i.e., each node in the MCTS tree of PromptAgent is now a complete reflection run. We provide the prompt of the checker in Fig. 4. During testing, we compared the performance of the LLM-based checker with a ground-truth checker, which is only available under specific dataset test conditions. The accuracy of the LLM-based checker is 89%. Although the 89% accuracy might seem reasonable, it's important to note that the success rate of a plan is less than 5%. This means that even a checker that consistently marks plans as incorrect would achieve better performance overall. We observe the same type of hallucinations as the ones in the plan generation, which is one of the main reasons that there are many false negatives in this checker. Meanwhile, we have briefly tested 5 of the false negatives with the latest GPT-4o, and we found that, surprisingly, they are all correct. However, to make a fair comparison and for the consistent of models used in the experiment, we are still using GPT-4-turbo as the model for the checker.

Additionally, we have made some necessary changes to the gradient descent prompt template to remove the requirement of labels, also known as the ground-truth answers, which in our case is also not provided.

To make the comparison between REPROMPT and PromptAgent fair, we use the Lite version of PromptAgent to limit the number of iterations. However, even with PromptAgent-Lite, it is still about twice as expensive compared to REPROMPT, which shows another advantage of our algorithm.

## B.2 Meeting Planning Dataset

> You are a format transformer. You will be provided with a meeting plan, your job is to transform it into a more strict output format that start with word "You" and then to more detailed activity. Here are some examples:
> You start at Russian Hill at 9:00AM.
> You travel to Marina District in 7 minutes and arrive at 9:07AM.
> You wait until 3:45PM.
> You meet James for 75 minutes from 3:45PM to 5:00PM.
> Try to change only the location, time, and name in your output for the above examples. Begin your final answer with: "The formatted output is: ".
> ==== Here is the solution you need to transform:
> Solution

Figure 3: The prompt used to transform the output from Deepseek-R1 from 0-shot prompt to the desired format. The prompt should be fed into an LLM, which in the experiment is Deepseek-V3 to transform the output to meet the desired format of Natural Plan.

The dataset is provided in the format of directly providing prompts in both 0-shot settings and 5-shot settings. However, the checker from the original repository only support a specific output format, that is inferred only in the 5-shot version of the prompt. Nevertheless, the official guideline of Deepseek-R1 has instructed users to always use 0-shot settings et al. (2025) as few-shot prompts always significantly degrade the performance. Considering all of these, we use the 0-shot prompt as the initial prompt, collect the results, and use Deepseek-v3 to transform the collected solution to the desired output format. It need to be noticed that this additional process may introduce certain responses that should be correct to go wrong. We provide the prompt of this transformation in Fig. 3.

## C  Additional Experiment Results

|             | GPT-4-turbo         | GPT-4o            |
|-------------|---------------------|-------------------|
| Reflexion   | 0.22, 0.11, 2.78    | 2.44, 5.67, 6.00  |
| PromptAgent | 0.11, 1.89, 2.11    | 5.67, 3.67, 2.56  |
| RePrompt    | 0.22, 3.44, 3.89    | 2.22, 7.67, 8.00  |

Table 5: Full list of pass rate results of GPT-4-turbo and GPT-4o on Travelplanner.

It needs to be highlighted that at the time of the experiments are conducted, even if the prompt, the temperature, the model used, and the seed are the same, OpenAI APIs still do not guarantee that the generated output will be exactly the same every time. This will greatly affect the final results in our case, given that REFLEXION (Shinn et al., 2023) is used to provide the feedback, and a small change on the earlier reflection can lead to completely different results in the end. Due to unknown reasons, we found that the reflection module in REFLEXION (Shinn et al., 2023) can provide completely useless and even wrong suggestions to the LLM and lead to very bad results. If this happens, we suggest rerunning the baseline model without RePrompt to make sure the OpenAI is providing correct feedbacks. However, the final conclusion, especially the superiority of our model, is always found to be true in our experiments. And for a fair comparison and to match the results from the TravelPlanner paper, all the results reported in the paper are the best among the 3 trials over time (Best-of-3 with same parameters). For reference, in Table 5, we provide the final pass rate of the 3 runs we have, together with additional results from GPT-4o. We observe that while our improvements are not statistical significant given the huge underlying random variations on the pass rate, the results are overall better when numbers are sorted in order.

> You are an AI assistant. Your job is to determine whether a specific travel plan meets the constraints of both commonsense constraints like reasonable route and hard constraints like budgets. The constraints are provided below. You will be provided with some information about the candidate trip contents, a specific query, and a proposed solution to the query. Please analyze the query and evaluate whether the query is correct or not. Note that all the information in the plan should be derived from the provided data, and do not do any additional estimation or approximation. Put your final judgment of "Correct" or "Wrong" in a \bbox{}.
>
>
> ==== Given information: {information}
> Query: {query}
> The generated plan is: {plan}

Figure 4: The checker prompt for PromptAgent. The prompt should be fed into an LLM, which in this paper is GPT4-turbo, to get the judgement of whether the current result is correct or wrong.

## D  Pseudo Code for RePrompt

While in the main paper, we only provided the workflow of our paper, here we provide actual pseudo code. While REPROMPT can be an analogy to fine-tuning on the prompt space, the code is very similar to a typical ML training loop as shown in Alg. 1.

---

**Algorithm 1** REPROMPT Train Loop

---

1: **function** TRAIN
2:     $Prompt \leftarrow$ Initial Prompt
3:     **for** (batch, x)  in Dataloader **do**
4:         $Response \leftarrow$ ACTOR($Prompt, x$)
5:         $Loss \leftarrow$ LLM_SUMMARIZER($Response, x, Prompt$)
6:         $Prompt \leftarrow$ PROMPTOPTIMIZER($Loss, Prompt$)
7:     **end for**
8:     **return** $Prompt$
9: **end function**

---

## E  Prompts for RePrompt

Here, we provide all the prompts used in our paper. The prompt used to summarize the loss is provided in Fig. 5, the prompt used to optimize the prompt is provided in Fig. 9, and the prompt used to replace the placeholders that could be accidently included, which is discussed in Sec. 5, is provided in Fig. 8. For prompt initiation, we first use a checker to check whether the prompt already include steps, whose prompt is provided in Fig. 6, and if not, we will call a prompt-formatter to add a two steps into the initial prompt, whose prompt is provided in Fig. 7. In our experiments, only the PDDL task has triggered this prompt-formatter. And given the popularity of CoT used in all reasoning-related tasks, which generally include all LLM-agent tasks, we believe these two LLMs are no longer needed for any new tasks.

## F  Optimized Prompt from RePrompt

Here, to help reproducibility, we provide all the optimized prompts that leads to the results shown in Table. 1 and Table. 2.

In Fig. 10, we provide the optimized prompt for PDDL action generation. In Fig. 11 and Fig. 13, we provide the optimized prompt for the TravelPlanner environment (Xie et al., 2024). In Fig. 12, we provide the

---

You are a summarizer. You wil be provided with a chat history from an AI assistant and the user. Please choose one of the following that you believe is the case, and summarize the focus point as instructed:
a). You can summarize the main reason for failures that led to this length of discussion. You only need to summarize the reason that has appeared, but not further summarize and infer the reason behind all the reasons. Make sure you choose only one reason at a time.
b). There is a specific thought or a list of similar thoughts that is very helpful to getting the correct answer. In this case, try to generalize the thought and make it does not involve detail information like concrete numbers, but as a high-level thought of what aspect should be highlighted and focus on.
c). There is no general reason that leads to a failure. It is case-by-case errors that is inevitable.
First, do some short analysis, and then finish your conclusion in one single line, starting with: "In conclusion, the main focus point should be: "
User Prompt
Here is the chat history, please follow the instructions above and tell me what is the main focus point should be in the required format:
<Chat History>

Figure 5: The loss summarize prompt. The prompt should be fed into an LLM, which in this paper is GPT4-turbo, to get the loss used to optimize the prompt.

---

You are a identifier. You will be given a prompt. You need to determine if the prompt has step-by-step instructions. If the prompt has step-by-step instructions, say True. Otherwise, say False. Do not provide any additional information.

---

Figure 6: The prompt to check whether the current prompt has already included multiple steps.

---

You are a prompt converter. You will be given a prompt. Your job is to convert the prompt into a step-by-step instruction format.
To do so, add an instruction before the examples part in the prompt. Add a sentence of "Let's do it step-by-step.
- Give some initial analysis.
- Provide the final answer."
Output directly the new prompt with step-by-step instructions. Do not provide any additional note or analysis.

---

Figure 7: The prompt to check whether the current prompt has already included multiple steps.

---

You are a template replacer. You will be provided with an original prompt, and an optimized prompt. Part of the new optimized prompt is a placeholder that needs to be replaced with the original prompt. Your job is to replace the placeholder with the original prompt.
One example of the placeholder is: " ⟨ Original Prompt Start ⟩ ". You need to replace this placeholder with the original prompt.
Another exmpale is <Examples from the original prompt>. You need to replace this placeholder with the examples from the original prompt.
Output directly the new prompt with the placeholder replaced. Do not provide any additional note or analysis.

---

Figure 8: The prompt to fix the place holders in the optimized prompt.

System Prompt

You are a prompt optimizer. You will be provided with an original prompt, and a specific point that this round of optimization should focus on. Your job is to update the prompt based on the provided focus point. If the focus point is saying there is no general reason, then skip all the following step and directly output the original prompt.

In the process, do the following steps one by one:

1. List a few different options that could address the given focus point.

2. Choose the solution that you think is the most promising. Make sure the solution is focus on instruction on how to solve the problem rather than instructions on giving better problem description. The solution should not be too general and should bring in actual insights.

3. Analyze each steps in the original prompt, and see whether the new solution should be inserted before or after the current step, or it is a superset of the current step and thus the original step should be replaced.

4. Finish your output with your final prompt, in the format of: "Based on the above analysis, the improved prompt is: ".

A few common solutions for specific problems are:

- If some details are missed, a sentence by sentence check ahead of time could be helpful.

- If some requirement are not meet, then a first analysis on that constraint could be helpful, or keep satisfying that requirement in mind when giving the solution could be useful.

- If it is already a thought, then a check on whether the thought is still workable in the given scenario is very helpful. For example, if it is about a speicific requirement need to be meet, then maybe also make sure to check it in every step. However, make sure this does not limit what the feedback can provide, and using words like "specifically" to remind such a check.

During the process, make sure that you focus on optimizing the prompt for the given focus point, and do not provide any additional information.

Do not change any other part of the prompt. Only focus on the step-by-step instructions. Especially, do not change the examples and the format requirement. However, make sure you copy the detailed previous example completely to the new output instead of using place holders to indicate that it should not be changed. Do not worry about the output length caused by the examples.

Please provide a detailed and complete response without omitting any information or use "..." or "[...]"to replace any part of the prompt. Again, ensure that no information is omitted or summarized.

Figure 9: The prompt optimizer prompt. The prompt should be fed into an LLM to update the prompt for problem solving.

You are defining the preconditions and effects (represented in PDDL format) of an AI agent's actions. Information about the AI agent will be provided in the domain description. Note that individual conditions in preconditions and effects should be listed separately. For example, "object_1 is washed and heated" should be considered as two separate conditions "object_1 is washed" and "object_1 is heated". Also, in PDDL, two predicates cannot have the same name even if they have different parameters. Each predicate in PDDL must have a unique name, and its parameters must be explicitly defined in the predicate definition. It is recommended to define predicate names in an intuitive and readable way. Here are two examples from the classical BlocksWorld domain for demonstrating the output format. <Examples from the original prompt>
Before defining the preconditions for an action, consider the implications of the action within the given domain. Identify any additional preconditions that are critical for the action to be performed successfully. Ensure that all necessary conditions are accounted for before listing them.
Here is the task.
Domain information: {domain_desc}
Action:

Figure 10: The optimized prompt for PDDL generation. The main changes are highlighted in blue.

You are a proficient planner. Based on the provided information and query, please give me a detailed plan, including specifics such as flight numbers (e.g., F0123456), restaurant names, and hotel names. Note that all the information in your plan should be derived from the provided data. You must adhere to the format given in the example. Additionally, all details should align with common sense. Attraction visits and meals are expected to be diverse. The symbol '-' indicates that information is unnecessary. For example, in the provided sample, you do not need to plan after returning to the departure city. When you travel to two cities in one day, you should note it in the 'Current City' section as in the example (i.e., from A to B). Before starting the planning process, establish a budget breakdown for each category (transportation, meals, attractions, accommodation) to ensure that the total cost does not exceed the provided budget. Solve this task by alternating between Thought, Action, and Observation steps. The 'Thought' phase involves reasoning about the current situation and specifically the budget constraints.
The 'Action' phase can be of two types:
(1) CostEnquiry[Sub Plan]: This function calculates the cost of a detailed sub plan, which you need to input the people number and plan in JSON format. The sub plan should encompass a complete one-day plan. An example will be provided for reference.
(2) Finish[Final Plan]: Use this function to indicate the completion of the task. You must submit a final, complete plan as an argument.
***** Example *****
<Examples>
***** Example Ends *****
{reflections}
You must use Finish to indict you have finished the task. And each action only calls one function once.
Given information: {text}
Query: {query}{scratchpad}

Figure 11: The prompt of TravelPlanner optimized after 1 epoch of REPROMPT. The main changes are highlighted in blue.

optimized prompt for Meeting Planning task. In Fig. 14, we provide the optimized prompt generated by PromptAgent Wang et al. (2024) for TravelPlanner environment.

You are visiting San Francisco for the day and want to meet as many friends as possible. Solve the problem by considering various different schedules and picking the best one to optimize your goals.

1. **Identify mandatory time constraints**: Start by listing all fixed commitments (e.g., travel to/from the city, meal breaks, pre-scheduled events). Calculate total time consumed by these.

2. **Maximize flexible availability**: Subtract mandatory time from your total available hours. This remaining time will be used for friend meetings.

3. **Evaluate scheduling options**:

a. **Identify friend clusters**:

- **First analyze proximity efficiency factors**:

- Evaluate mandatory commitments' locations **and time windows**

- Identify transit corridors/routes connecting multiple commitments

- Map time-sensitive opportunities (e.g., friends available during/after nearby commitments)

- Form initial clusters by:

- Grouping friends within 15-minute transit of **any mandatory commitment location or transit corridor**

- Prioritizing clusters that align with commitment time windows (e.g., friends near lunch spot during meal break)

- Then map remaining friends into geographic clusters based on density

- Calculate transit time between all clusters and to/from mandatory commitments

b. Prioritize clusters that:

- **Use a scoring system**:

- Calculate cluster scores by combining:

- **Proximity score**:

- 8 points if ≤10min transit to nearest commitment/corridor

- 5 points if 11-20min

- 2 points if 21-30min

- 0 points if >30min

- **Density score**: (Number of friends in cluster) × (3 ÷ cluster radius in miles)

- Example: 6 friends in 0.6-mile radius = 6 × (3/0.6) = 30 points

- Prioritize clusters with the highest combined scores first

- **Re-evaluate scores after completing each mandatory commitment** to account for new proximity opportunities

- **Structure clusters to**:

- Start with clusters **adjacent to early mandatory commitments** to establish proximity early

- **After completing any mandatory commitment, remain in its cluster** to meet nearby friends and avoid backtracking

- Sequence clusters **along transit corridors in one direction** (e.g., north-to-south) to minimize crisscrossing

- Place clusters scoring ≥35 total points **immediately after mandatory commitments** when time windows allow extended stays

- Enable efficient sequential visitation with minimal internal transit time

c. Compare scenarios where you:

- Start early vs. late

- Sequence clusters by proximity to mandatory commitments first

- Adjust meeting durations within clusters

4. **Select the optimal schedule**: Choose the option that maximizes the number of friends met while respecting all constraints.

query

Your response should start with 'SOLUTION:'.

Figure 12: The prompt of Meeting Planning optimized after 5 epoch of REPROMPT. The main changes are highlighted in blue.

You are a proficient planner. Based on the provided information and query, please give me a detailed plan, including specifics such as flight numbers (e.g., F0123456), restaurant names, and hotel names. Before you start planning, conduct a preliminary budget analysis to understand the cost constraints for each category (transportation, accommodation, meals, and attractions). Ensure that the accommodation information is formatted according to the predefined template compatible with the cost calculation environment. After setting the preliminary budget, conduct a comparative analysis of transportation options to select the most cost-effective one, research meal options to find the best value that fits dietary preferences and proximity requirements, and compare accommodation choices based on cost, location, amenities, and reviews. Set specific budget limits for meals and accommodations to ensure the overall expenses do not exceed the budget while maintaining a satisfactory experience. Ensure that each choice of transportation, accommodation, meal, and attraction is tailored to the specific preferences and requirements provided in the query, making iterative adjustments to the plan as necessary to stay within budget constraints. Note that all the information in your plan should be derived from the provided data. You must adhere to the format given in the example. Additionally, all details should align with common sense. Attraction visits and meals are expected to be diverse. The symbol '-' indicates that information is unnecessary. For example, in the provided sample, you do not need to plan after returning to the departure city. When you travel to two cities in one day, you should note it in the 'Current City' section as in the example (i.e., from A to B). Before starting the planning process, establish a budget breakdown for each category (transportation, meals, attractions, accommodation) to ensure that the total cost does not exceed the provided budget. Solve this task by alternating between Thought, Action, and Observation steps. The 'Thought' phase involves reasoning about the current situation and specifically the budget constraints. The 'Action' phase can be of two types: (1) CostEnquiry[Sub Plan]: This function calculates the cost of a detailed sub plan, which you need to input the people number and plan in JSON format. The sub plan should encompass a complete one-day plan. An example will be provided for reference. (2) Finish[Final Plan]: Use this function to indicate the completion of the task. You must submit a final, complete plan as an argument.
***** Example *****
<Examples>
***** Example Ends *****
{reflections}
You must use Finish to indict you have finished the task. And each action only calls one function once.
Given information: {text}
Query: {query}{scratchpad}

Figure 13: The prompt of TravelPlanner optimized after 5 epochs of REPROMPT. The main changes are highlighted in blue.

You are a proficient planner with a keen eye for detail and practicality. Your task is to create a comprehensive travel plan that adheres to the provided budget and timeframe, ensuring a diverse and enjoyable experience. The plan should include specific flight numbers, restaurant names, hotel names, and attraction details, all of which must be derived from the provided data. Follow the format shown in the example, and ensure that all details are sensible and feasible.

When planning, consider the following guidelines:

- Ensure meal diversity by not repeating restaurant choices for different meals.
- Select transportation options that are practical and feasible, considering the distance and geography.
- Provide complete information, including all meals for each day and any in-city transportation if necessary.
- Adhere to the budget, allocating funds across flights, accommodations, meals, and attractions.
- Check for any minimum stay requirements or house rules for accommodations.
- Use the symbol '-' to indicate when information is unnecessary, such as after returning to the departure city or when no transportation is needed within the current city.

Your planning process should consist of alternating Thought, Action, and Observation steps:

- Thought: Reason about the current situation and what needs to be planned next.
- Action: Perform one of two types of actions: (1) CostEnquiry[Sub Plan]: Calculate the cost of a detailed sub-plan for a complete one-day plan. Input the number of people and the plan in JSON format.
(2) Finish[Final Plan]: Indicate the completion of the task by submitting a final, complete plan as an argument.
- Observation: Reflect on the information received from the actions and adjust the plan accordingly.

Remember, each action should only call one function once, and you must use Finish to indicate you have finished the task.

Here is an example for reference:

***** Example *****

{Example content}

***** Example Ends *****

Now, let's begin planning based on the given information and query. Keep in mind that the plan should be logical, feasible, and within the specified constraints. Good luck!

{reflections}

Given information: {text} Query: {query}{scratchpad}

Figure 14: The prompt of TravelPlanner optimized by PromptAgent. Unlike REPROMPT, majority prompt has been changed and thus we do not do further highlight.

