# OpenReview forum: "RePrompt: Reflection-based Automatic Prompting for LLM Agents"
_TMLR — Rejected by TMLR_

### Review · Reviewer_FwTH · 2025-06-15

**Summary Of Contributions:**

This paper presents RePrompt, an automatic prompt optimization method designed for reasoning tasks in large language model (LLM) agents. The approach utilizes intermediate feedback from interactions with LLM agents to refine prompts in a manner similar to gradient descent. Unlike prior automatic prompt engineering (APE) techniques that rely on final solution checkers, RePrompt leverages step-by-step dialogue history to improve prompts iteratively. The method is evaluated across three tasks — PDDL generation, TravelPlanner, and Meeting Planning—demonstrating improvements in prompt effectiveness.

**Audience:**

Yes

**Claims And Evidence:**

Yes

**Requested Changes:**

Discussion on similar approaches (such as OPRO and TextGrad) and experimental comparisons with them should be included.

**Strengths And Weaknesses:**

## Strengths

* The proposed framework is technically sound. It is well-motivated and grounded in existing approaches like Chain-of-Thought prompting, Reflexion, and ReAct. The method logically extends previous work to optimize prompts without requiring ground-truth evaluators.
* The presentation is clear. The paper is structured effectively, with a clear problem statement, methodology, and experimental validation.

## Weaknesses

* Missing discussion of similar methods. What are the key differences between the proposed method and OPRO (https://arxiv.org/pdf/2309.03409) or TextGrad (https://arxiv.org/pdf/2406.07496)? Especially, TextGrad also takes an analogy of the prompt optimization to gradient descent, and implements components such as loss and optimizer with LLMs.

---

> ### Author Response · Authors · 2025-06-23
>
> Thank you for your appreciation of our work.
>
> Regarding your concerns about the discussion of related approaches, we would like to emphasize that one of our main contributions is the extension of prompt optimization to a new test-time setting, as you also noted in the strengths section. Due to the fundamental shift in the problem formulation, direct experimental comparisons with prior methods are not feasible. We have expanded on this point in both our general response and the updated version of the paper.
>
> We invite you to review these revisions, and we would be happy to address any remaining concerns you may have.

---

### Review · Reviewer_XhSh · 2025-06-16

**Summary Of Contributions:**

This paper proposes RePrompt, an automatic prompt optimization framework tailored for LLM agents in reasoning tasks, particularly when ground-truth final checkers are unavailable or expensive. Instead of relying on explicit success/failure labels, RePrompt uses intermediate feedback from interactions (e.g., via ReAct or Reflexion) to iteratively refine the step-by-step instructions in prompts. The method mimics a gradient-descent-like process, using a summarizer to identify recurring bottlenecks in chat histories and a prompt optimizer to improve instruction granularity and relevance.

**Audience:**

Yes

**Broader Impact Concerns:**

No ethics problems involved.

**Claims And Evidence:**

Yes

**Requested Changes:**

1. As seem above, the authors should first clarify their novelty and provide more experiments
2. The authors should polish the paper. Expressions like "In this paper, we not only have human domain experts but also use another LLM as a separate checker to verify whether any of the errors that appear in the results given by the prompt in the original paper (Guan et al.,
2023), which are released together with the code of it, have also appeared in our result." is hard to be understood.

**Strengths And Weaknesses:**

Strengths:
1. Automaticity: The method is an automatic method that doesn't require humans to refine prompts.
2. Interesting Error Analysis

Weakness:
1. Lack of novelty: The authors should clarify the difference between the paper and TextGrad [1].
2. Insufficient experiments and baselines: The authors should compare their methods with more baselines, including prompting techniques (CoT, Reflexion...). The authors should also try more LLMs backbones.
3. Convergence: It seems hard to judge whether the prompt has converged automatically.

[1] TextGrad: Automatic "Differentiation" via Text

---

> ### Author Response · Authors · 2025-06-23
>
> Dear reviewer,
>
>
> Thank you for your time on our paper, here we address your concerns one by one:
>
>
> > lack of novelty
>
>
> As you noted in your summary, our work explores a new test-time prompt optimization setting where ground-truth final checkers are either unavailable or costly to obtain. A key contribution of our paper lies in extending existing prompt optimization frameworks to this more challenging setting. In response to this shift, we also modify the algorithm to focus on optimizing the intermediate steps within the prompt, which is critical when final supervision signals are limited or missing. We have revised the paper to better highlight the novelty of this setting and, in our general response, we draw an analogy between our work and the application of reinforcement learning in large language models. We invite you to also review the general response and our updated version of the paper for further clarification.
>
>
> > The authors should compare their methods with more baselines, including prompting techniques
>
>
> As you noted in the strengths, our work focuses on automatic prompt optimization. Since there is no inherent limitation on the prompt space accessible to automatic methods compared to manual prompt engineering, our primary goal is to demonstrate improvements over the base prompts provided by existing datasets. Regarding specific techniques such as chain-of-thought and Reflexion, we note that Reflexion is already included as a baseline in the TravelPlanner task. Additionally, chain-of-thought prompting is no longer effective for models like Deepseek R1 even if it is also already included in the prompt, which is used in the NaturalPlan experiments. In both settings, RePrompt demonstrates clear improvements over the respective base prompts.
>
>
> > The authors should also try more LLMs backbones.
>
>
> Thank you for your suggestion. We have added results using GPT-4o on the TravelPlanner benchmark in the appendix.
> In addition to these updates, we have revised the paper to improve clarity and incorporate feedback from the other reviewers. We invite you to review the updated version, and we would be happy to address any remaining concerns you may have.

---

> > ### Comment · Reviewer_XhSh · 2025-07-08
> >
> > Thanks to the author for clarification.
> > 1. I'm not convinced by the test-time statement. The concept of "epoch" is involved in the experiments, which is irrelevant to the test time. If "epoch" is involved, the difference between this paper and textgrad lies in the designation of the objective function (using LLM as a judge vs using a verifier). This discrepancy can't convince me that the novelty is enough.
> >
> > 2. The proposed method needs several epochs to improve the performance. However, at test time, each case is tested for once. Furthermore, when the proposed method is only performed for 1 epoch, the performance doesn't surpass traditional prompting techniques in all aspects.
> >
> > 3. Thank the author for adding experiments on using more LLM backbones.

---

> > > ### Author Response · Authors · 2025-07-08
> > > **Further Clarification on Test-time Statement**
> > >
> > > Dear Reviewer XhSh,
> > >
> > > Thank you for your follow-up response.
> > >
> > > We understand that test-time reinforcement learning may be outside your primary area of focus, and you may not have reviewed that work in detail. We would therefore like to offer further clarification regarding the concept of "test-time" as it relates to points 1 and 2 in your latest comments.
> > >
> > > In both our work and in test-time reinforcement learning [1], the term "test-time" refers to scenarios where final ground-truth labels are unavailable. It does not imply that “each case is tested for once”. Similar to test-time reinforcement learning, our approach involves multiple epochs during inference to iteratively improve performance. In fact, test-time reinforcement learning goes even further to updating model parameters for multiple epochs. The definition of the objective function under these constraints is what drives the need for novel algorithmic design. In our case, we would like to emphasize the additional novelty of explicitly optimizing the intermediate steps in prompts, which is essential in the absence of final supervision.
> > >
> > > Regarding your concern about performance after a single epoch, we would like to clarify that our method is not designed to converge within one epoch. Without access to ground-truth labels, and relying solely on noisy feedback histories, it is extremely difficult to make meaningful updates within a single epoch. However, our experiments on the PDDL generation task serve as a special case where feedback is highly accurate, and in this scenario, our method is able to perform well even after just one epoch.
> > >
> > > Please let us know if you have any further questions or concerns. We would be happy to continue the discussion.
> > >
> > > [1]. Zuo Y, Zhang K, Sheng L, et al. Ttrl: Test-time reinforcement learning[J]. arXiv preprint arXiv:2504.16084, 2025.

---

> > > > ### Comment · Reviewer_XhSh · 2025-07-10
> > > >
> > > > Thanks for your clarification. I believe we have reached a common ground on the novelty when compared with TextGrad lies in Test-Time. If the authors want to change the story to test time, the introduction should be rewritten to adapt to the new theme. Furthermore, if test time is the new story, test-time baselines, such as Best-of-N, and so on,  should be discussed and compared.
> > > >
> > > > Apart from the above, TTRL [1] isn't published by any conference, and it's questioned by the problem of using testing data as training data, violating the train-test split principle. The authors train with batches on the testing benchmark with more than 1 epoch, then this problem should be clarified.
> > > >
> > > > I understand that the method performs better after just one epoch. However, what I'm trying to highlight is that if the authors cannot address the issue of the train-test split or justify the use of multiple epochs in the testing benchmark, which may stuck the new story of test-time, they might want to reframe their new contribution as a streaming approach. That said, I would like to point out upfront that the streaming version, i.e., the one-epoch version, yields unsatisfactory results.
> > > >
> > > > [1] Ttrl: Test-time reinforcement learning

---

> > > > > ### Author Response · Authors · 2025-07-10
> > > > > **Further Response**
> > > > >
> > > > > Dear Reviewer Xhsh,
> > > > >
> > > > >
> > > > > Thanks for your further response. Here we address your concerns one by one:
> > > > >
> > > > >
> > > > > >  I believe we have reached a common ground on the novelty when compared with TextGrad lies in Test-Time.
> > > > >
> > > > >
> > > > > We agree that the primary contribution lies in the novelty of the setting. However, we would like to emphasize that on the methodological side, RePrompt also introduces an additional contribution by explicitly optimizing the intermediate steps within prompts, which distinguishes it from TextGrad and other existing automatic prompt engineering approaches.
> > > > >
> > > > >
> > > > > >  If the authors want to change the story to test time, the introduction should be rewritten to adapt to the new theme.
> > > > >
> > > > >
> > > > > Thank you for your suggestion. However, we believe that the central theme of our work has consistently focused on the question: *How can automatic prompt engineering be applied when no solution checker is available for the task?* During the rebuttal and discussion phase, we recognized that the term "test-time" precisely captures this setting, and we therefore decided to incorporate it into the paper. The current introduction still clearly outlines the challenges of applying existing automatic prompt engineering methods to such problems, which directly motivates our proposed solution.
> > > > > That said, we have made a minor revision to the introduction to further clarify this point. Again, the changes are highlighted in red, and we invite you to review them.
> > > > >
> > > > >
> > > > >
> > > > >
> > > > > > Furthermore, if test time is the new story, test-time baselines, such as Best-of-N, and so on, should be discussed and compared.
> > > > >
> > > > >
> > > > > We believe there may be a misunderstanding between the concepts of “test-time” and “test-time scaling”. In this paper, our focus is specifically on test-time prompt optimization, not on test-time scaling strategies. As demonstrated in our experiments, we can optimize the prompt using a small set of examples, such as 10 data points in the TravelPlanning and MeetingPlanning tasks, and then apply the optimized prompt to a larger test set. In contrast, best-of-N is a test-time scaling approach that requires additional computation for each individual input by generating multiple responses. Since the objectives and resource requirements of these two settings are fundamentally different, they are not directly comparable.
> > > > >
> > > > >
> > > > > >  TTRL [1] isn't published by any conference, it's questioned by the problem of using testing data as training data, violating the train-test split principle
> > > > >
> > > > >
> > > > > We respectfully disagree with the claim that our work violates the train–test split principle. This principle is intended to prevent the use of **test labels** during training, not the use of **test data**. Both in test-time reinforcement learning (TTRL) and in our work, test data may be partially accessed, but no test labels are ever used. Therefore, our approach remains consistent with the core requirement of the train–test split.
> > > > >
> > > > >
> > > > > > The authors train with batches on the testing benchmark with more than 1 epoch, then this problem should be clarified.
> > > > >
> > > > >
> > > > > We sincerely apologize, but we are unclear about the rationale behind the claim that training should be limited to only one epoch. If this concern is based on the earlier point regarding the train and test split principle, then by that logic, no training at all, not even a single epoch, should be permitted in order to strictly adhere to that principle. However, as we previously explained, the purpose of the train and test split principle is to prevent the leakage of test labels, not test data. Since our approach does not use any test labels, training over multiple epochs should be entirely acceptable within this framework.
> > > > >
> > > > >
> > > > > Thank you again for your detailed comments and for dedicating your valuable time on our paper. We would be happy to address any further questions or concerns you may have.

---

> > > > > > ### Comment · Reviewer_XhSh · 2025-07-11
> > > > > >
> > > > > > Since you agree that the story has changed to test time, you should rewrite the whole paper to justify it. The current version is strange, such as in the introduction part, the motivation for using test-time should be justified, and related works should include test-time optimization methods. Also, your insights should be related to test-time settings.
> > > > > >
> > > > > > Best-of-N can be a test-time scaling method, but you can set N=3or4, which doesn't lie in a large scale. If you change the story to test-time, test-time methods should be respected.
> > > > > >
> > > > > > For the train-test split principle, in the RL settings, which are highly similar to yours, no label is offered, but generalization to an unseen environment is needed.
> > > > > >
> > > > > > > We sincerely apologize, but we are unclear about the rationale behind the claim that training should be limited to only one epoch.
> > > > > >
> > > > > > If the training is limited to 1 epoch, then it's a streaming setting.

---

> > > > > > > ### Author Response · Authors · 2025-07-11
> > > > > > > **Further Further Response to Reviewer Xhsh**
> > > > > > >
> > > > > > > Dear Reviewer Xhsh,
> > > > > > >
> > > > > > >
> > > > > > > Thank you for your further response again. Here we address your remaining concerns one by one:
> > > > > > >
> > > > > > >
> > > > > > > > Since you agree that the story has changed to test time
> > > > > > >
> > > > > > >
> > > > > > > As we noted earlier, the story of our work remains consistent before and after the rebuttal. The only change is in the terminology, where we introduced the term "test time" to more accurately describe the problem setting. Our contribution remains unchanged: with the proposed algorithm RePrompt, we demonstrate that prompt optimization can be performed at test time without requiring access to a ground-truth checker. Therefore, we do not think there is any need to rewrite the entire paper.
> > > > > > >
> > > > > > >
> > > > > > > > For the train-test split principle, in the RL settings, which are highly similar to yours, no label is offered, but generalization to an unseen environment is needed.
> > > > > > >
> > > > > > >
> > > > > > > In our experiments on the TravelPlanner and MeetingPlanning tasks, we perform prompt optimization on a small subset of the test set and then evaluate the optimized prompt on the full test set. Specifically, for TravelPlanner, we optimize the prompt for five epochs using 10 data points, and then report the pass rate on 180 data points. For the MeetingPlanning task from NaturalPlan, we optimize the prompt for five epochs using 9 data points, and evaluate on 100 data points. In both cases, the evaluation setup already includes a generalization component, while ensuring that no test labels are used during the optimization process.
> > > > > > >
> > > > > > >
> > > > > > > > Best-of-N can be a test-time scaling method, but you can set N=3or4, which doesn't lie in a large scale.
> > > > > > >
> > > > > > > We would like to reiterate that the concept of "test time" is fundamentally different from "test time scaling." In fact, the former can be seen as a much broader category that encompasses a wider setting, whereas the latter refers to a specific class of methods. In our case, the newly introduced test time prompt optimization setting is not directly comparable to test time scaling approaches at all.
> > > > > > >
> > > > > > > In terms of computational cost, our method introduces only a 28 percent overhead on the TravelPlanner task and a 45 percent overhead on the NaturalPlan task. This is significantly lower than a Best-of-N strategy with N equal to 2, which incurs a 100 percent increase in cost. Moreover, the cost efficiency of our approach becomes even more pronounced as the optimized prompt is reused across a larger number of test instances.
> > > > > > >
> > > > > > >
> > > > > > > > If the training is limited to 1 epoch, then it's a streaming setting.
> > > > > > >
> > > > > > >
> > > > > > > Thank you for your clarification. We hope our previous response has sufficiently addressed why the constraint of limiting training to 1 epoch is not necessary in our setting.

---

### Review · Reviewer_zJcF · 2025-06-16

**Summary Of Contributions:**

The paper proposes RePrompt, new automatic prompt tuning method seeking to optimize the output of LLM systems. The method specifically targets LLM systems that use iterative feedback to work towards a final solution, where the iterative feedback could come from an automated feedback generator such as ReAct, or from humans. The method consists of two parts: (1) An LLM-based "summarizer" takes a batch of outputs from the system (including feedback) and summarizes the main issue identified in the iterative feedback. (2) An LLM-based "prompt optimizer" updates the initial prompt provided to the LLM system based on the summarized issue form the summarizer. Intuitively, the idea is not necessarily to improve the initial prompt with ground-truth data (which most generated feedback is not), but rather to integrate common feedback directly into the prompt, and thereby improve the initial output of the LLM. This process can be repeated several times.

RePrompt is evaluated on three benchmarks (PDDL model construction, TravelPlanner and Meeting Planner), using GPT-4-turbo as the basis for the LLM system, the summarizer and the prompt optimizer. In each benchmark a different method for generating feedback is used. The authors show that in all three benchmarks, RePrompt helps the LLM generate better solutions than a baseline LLM.

**Audience:**

Yes

**Broader Impact Concerns:**

No broader impact statement is present. However, I do not see any concerns beyond those common to the wide body of LLM research that would need to be raised specifically.

**Claims And Evidence:**

Yes

**Requested Changes:**

Addressing the two weaknesses I raised above I would be critical in securing my recommendation for acceptance:
- Please report on the statistical significance of key results in Tables 1-3.
- If feasible, please include results on other LLM models.

Requests for clarification:
- Section 3.2.3 explains that if there is no step-by-step instructions present in the initial prompt, these are added before RePrompt is run by an auxiliary checker. However, I could not find any detail on how this checker works, on which use cases it was used, and what its impact was on the results. Could you please add this information or point me to it in the paper?
- Appendix E says "for a fair comparison and to match the results from the TravelPlanner paper, all the results reported in the paper are the best among the 3 trials over time (Best-of-3).” Some clarification here is needed as it is not clear to me which trials are referred to here, or what the differences were between these trial.

**Strengths And Weaknesses:**

Strengths:
- RePrompt is easy to implement, making it an accessible method to improve most LLM agents.
- RePrompt achieves improvements in answer quality in all domains tested.
- The method is described in sufficient detail and all prompts used for the experiments (as well as other experiment details) are provided, making the results easy to reproduce.
- Explaining the rationale behind the construction of the prompts used is much appreciated.

Weaknesses:
- There are no tests for statistical significance on any of the results presented. For example, regarding the results in table 1, is the difference in the number of incorrect actions or the number of errors significant? Does RePrompt produce (statistically) significantly fewer errors per instance than the original results from Guan et al.? Similarly, are the differences in Table 2 and Table 3 significant?
- All experiments use GPT-4-turbo. This only really shows us that RePrompt works with this specific LLM. This is in contrast to prior work, for example the TravelAgent paper cited here, where several LLMs are tested (Mistral, Mixtral, Gemini Pro, GPT 3.5 and GPT 4 in that case). Testing on multiple LLMs would show that RePrompt is indeed a general method that can improve any LLM output rather than specifically GPT-4-turbo.

---

> ### Author Response · Authors · 2025-06-23
>
> Thank you for your detailed review. We are pleased that you found our descriptions clear and thorough. Below, we address your concerns point by point.
>
>
> > Please report on the statistical significance of key results in Tables 1-3.
>
>
> Unfortunately, due to the high cost of human evaluation for PDDL code, both in terms of time and monetary resources, we were not able to conduct a sufficient number of evaluations on the PDDL generation task to support statistical significance. This limitation prevents us from providing additional results for Table 1.
> For Tables 2 and 3, this also relates to your request for clarification on Appendix E, which has now been moved to Appendix C for improved readability. The three trials were conducted using exactly the same parameters. However, instead of reporting the mean and standard deviation, which are commonly used to assess statistical significance, we report the maximum performance among the three runs. As explained in the updated appendix, this decision was made because the TravelPlanner task exhibits significant variation on the final numbers, even when the same parameters are used in the original codebase without RePrompt. We have now included the complete set of pass rate results in the appendix. While the results are not statistically significant due to the high level of randomness, our method consistently shows improved performance when this variability is considered.
>
>
> > If feasible, please include results on other LLM models.
>
>
> In addition to the full list of pass rate results for GPT-4 Turbo provided in Appendix C, we have conducted further experiments using GPT-4o. The complete set of pass rate results for GPT-4o is also included in Appendix C. We observe that RePrompt continues to improve performance through prompt optimization. We would also like to highlight that our original paper already includes results using Deepseek R1 on the MeetingPlanning task, which is another set of results that demonstrates that our method is applicable to models other than GPT-4 Turbo.
>
>
> > Section 3.2.3 explains that if there is no step-by-step instructions present in the initial prompt, these are added before RePrompt is run by an auxiliary checker.
>
>
> Thank you for pointing this out. We have now included the corresponding prompts in the figure in the appendix. However, we would like to clarify that this is a one-time modification applied during initialization, which can also be performed manually. In our experiments, this mechanism was only triggered once in the PDDL generation task, as the initial prompt was already designed with step-by-step reasoning in mind. Given that chain-of-thought prompting is already widely used in reasoning-related tasks, we believe this additional checker is unlikely to offer substantial benefit for new tasks.

---

### Author Response · Authors · 2025-06-23
**General Responses to Reviewers Regarding Novelty and Updated Paper**

Dear reviewers,

Thank you for your time and valuable feedback on our paper. We are pleased that reviewers zJcF and FwTH found our approach to be sound and our description to be detailed. A recurring question concerns the distinction between our proposed algorithm, RePrompt, and existing methods such as TextGrad. We would like to clarify that one of our key contributions lies in extending prompt optimization to settings where a solution checker is not available.

We view our contribution as analogous to the extension of reinforcement learning (RL) in large language models (LLMs) to the test-time RL setting [1]. In both cases, the objective is to improve LLM performance when explicit external feedback is unavailable. While the high-level goals are similar to those in standard settings, both our work and the test-time RL extension require careful algorithmic redesign to address the unique challenges introduced at test time. This necessity arises not from differences between the two domains, but from the increased difficulty of optimization when direct supervision or solution checking is not accessible. In our case, the absence of a solution checker makes it difficult to identify which prompts lead to better outcomes, rendering the optimization direction substantially more ambiguous than in settings where a solution checker is present, such as in PromptAgent [2] and TextGrad [3]. To address this, our method aggregates feedback based on the history of generated actions, and selectively optimizes the steps in the prompt to enhance planning ability during the generation process.

To better illustrate the differences in problem setting, we have revised the manuscript accordingly, with the modifications clearly marked in red. We encourage you to review these updates.

Additionally, we have included new results in the appendix using GPT-4o on the TravelPlanning task. We find that RePrompt continues to yield performance improvements by optimizing the prompt, demonstrating its generalizability to different models.

[1]. Zuo Y, Zhang K, Sheng L, et al. Ttrl: Test-time reinforcement learning[J]. arXiv preprint arXiv:2504.16084, 2025.

[2]. Wang X, Li C, Wang Z, et al. Promptagent: Strategic planning with language models enables expert-level prompt optimization[J]. arXiv preprint arXiv:2310.16427, 2023.

[3]. Yuksekgonul M, Bianchi F, Boen J, et al. Textgrad: Automatic" differentiation" via text[J]. arXiv preprint arXiv:2406.07496, 2024.

---

### Decision · Action_Editor_A3qV · 2025-07-30

**Recommendation:** Reject

**Additional Comments:**

Most of my comments are above. Some additional notes here:

- Several reviewers raised concerns about novelty. While novelty on its own is not a criterion for TMLR acceptance, some of the concerns are indeed pertinent to the relation of the current work with respect to the literature, which will be relevant to the "Claims and Evidence" criterion. I encourage the authors to consider some of the points raised by the reviewers more carefully.

- Some reviewers are not happy with the change in narrative. Overall, the revision during rebuttal is meant for the authors to answer specific questions and address concerns through revising the paper in a limited way. A major revision / resubmission will be more appropriate if the authors believe that a larger-scale change in narrative is needed.

**Audience:**

Yes

**Audience Explanation:**

All reviewers agreed that this work would be of interest to some members of the TMLR audience; the AE concurs with this view.

**Claims And Evidence:**

No

**Claims Explanation:**

While all reviewers gave a "Yes" rating to this criterion in the original review, all of them determined that this paper falls short of this criterion in their final recommendation. Some of the important points raised are 1) the lack of comparison to baseline methods; 2) lack of statistical tests, and 3) the lack of sufficient experimental validation in multiple LLMs.

Regarding 1), authors emphasized on their "test-time setting" and that baselines require verification with ground-truth etc, an argument that did not sufficiently convinced the reviewers. Regarding this point, I'd also like to add some of my thought as well: firstly, the so-called "test-time setting" is not new c.f. [1-3]; some of these works optimize examples instead of instructions and use different names (e.g., transductive zero-shot), but fall in the same purview of prompt optimization without ground-truth labels, so the authors should tone down their claim and acknowledge these works. Second, I agree with the reviewers that the difference between this work and some of the previous works is not that large. For example, Sec 3.2 of TextGrad also features a usecase where ground-truth is not required and [4] is another work without using labels -- in general, I believe it is relatively easy to adapt existing works as comparable baseline, which could also act as an effective ablation study to emphasize the merits of the proposed method. As such, I'd agree with the reviewers that more experimental validation is required to demonstrate the better performance of the proposed algorithm rather than arguing that no suitable baseline exists.

For 2 and 3, I do sympathize with the authors' constraints but I still believe that some level of rigor is necessary especially given the authors mentioned themselves about the inherent randomness in LLMs. I encourage the authors to address these in a follow-up revision to alleviate concerns on experimental robustness. I also hope the time allowed for a more thorough revision will give authors more time in running the experiments.

References

[1] Wan, X., Sun, R., Dai, H., Arik, S. O., & Pfister, T. (2023). Better zero-shot reasoning with self-adaptive prompting. ACL.

[2] Wan, X., Sun, R., Nakhost, H., Dai, H., Eisenschlos, J. M., Arik, S. O., & Pfister, T. (2023). Universal self-adaptive prompting. EMNLP.

[3] Lyu, X., Min, S., Beltagy, I., Zettlemoyer, L., & Hajishirzi, H. (2022). Z-icl: Zero-shot in-context learning with pseudo-demonstrations. ACL.

[4]  Xiang, J., Zhang, J., Yu, Z., Teng, F., Tu, J., Liang, X., ... & Luo, Y. (2025). Self-supervised prompt optimization. arXiv preprint arXiv:2502.06855.

**Resubmission Of Major Revision:**

The authors may consider submitting a major revision at a later time.